# Autoencoder-Based General-Purpose Representation Learning for Entity Embedding

## Abstract

Recent advances in representation learning have successfully leveraged the underlying domain-specific structure of data across various fields. However, representing diverse and complex entities stored in tabular format within a latent space remains challenging. In this paper, we introduce DEEPCAE, a novel method for calculating the regularization term for multi-layer contractive autoencoders (CAEs). Additionally, we formalize a general-purpose entity embedding framework and use it to empirically show that DEEPCAE outperforms all other tested autoencoder variants in both reconstruction performance and downstream prediction performance. Notably, when compared to a stacked CAE across 13 datasets, DEEPCAE achieves a 34% improvement in reconstruction error.

## 1 Introduction

The underlying structure of data can be defined by its organization and relationships, encompassing semantic meaning, spatial positioning, and temporal sequencing across a range of domains. Lately, this structure has been leveraged to build use case-agnostic data representations in a self-supervised, auto-regressive, or augmentative manner (Pennington et al., 2014; Devlin et al., 2018; Liu et al., 2019; Conneau et al., 2019; Caron et al., 2020; Oord et al., 2018; He et al., 2020; Higgins et al., 2016), including the prediction of next tokens given the previous ones in GPTs (OpenAI, 2023), and image rotation (Gidaris et al., 2018).

Real world entities, such as products and customers of a company, generally stored in tabular format, can also be represented as multi-dimensional vectors, or *embeddings*. The demand for reusable entity embeddings is growing across research and business units, as a universally applicable representation for each entity can drastically reduce pre-processing efforts for a wide range of predictive models. This, in turn, can shorten development cycles and potentially enhance predictive performance. For example, customer embeddings could greatly improve sales and marketing analyses in large enterprises.

In industry settings, data pre-processing and feature engineering steps can constitute a large portion of a project's lifespan and result in duplicated efforts: Munson (2012) and Press (2016) showed that the average estimated percentage of time spent by data scientists for retrieving, pre-processing and feature-engineering data is between 50% and 70% of a data science project's lifespan. Furthermore, scientists may encounter challenges in feature selection and in identifying relationships within the data (e.g., correlation and causality). Complexity - frequently arising from high-dimensionality - can result in both overfitting and underfitting. Holistically, projects with different objectives relying on similar data could benefit from a unified shared pre-processing and feature engineering process, which would produce pre-processed data for a *general purpose*.

While text and images are structurally organized (syntax and semantics in text, spatial structure in images), tabular data does not necessarily exhibit such relations between features that could be leveraged for modeling purposes. Although many recent modality-specific representation learning methods (e.g. Transformers for text (Vaswani et al., 2017)) leverage the increase in computational capabilities, classical representation learning applicable to tabular data has yet to be advanced.

In this work, (1) we propose DEEPCAE, which extends the contractive autoencoder (CAE) framework (Rifai et al., 2011b) to the multi-layer setting while preserving the original regularization design, unlike stacked CAE approaches; (2) we outline a general-purpose end-to-end entity embed-

ding framework applicable to a variety of domains and different embedding models. We use the proposed framework to evaluate different representation learning methods across 13 publicly available classification and regression datasets, covering a multitude of entities (A.1). We measure both the reconstruction error as well as the downstream performance on the dataset's respective classification or regression task when using the embeddings as input. We show that DEEPCAE performs especially well in entity embedding settings (see Figure 2).

Our original contribution is the extension of the mathematical simplification introduced by Rifai et al. (2011b) for the calculation of the Jacobian of the entire encoder in the contractive loss from single-layer to multi-layer settings (see Section 4). This extension makes training the entire multi-layer encoder computationally feasible, allowing for increased degrees of freedom, while maintaining the benefits of regularization.

## 2 PRELIMINARIES

*Autoencoders* are a specific type of neural network designed for unsupervised learning tasks, whose main purpose is to encode inputs into a condensed representation, and are often used for dimensionality reduction and feature learning. The usually lower-dimensional space in which the input is projected is referred to as *latent space*, or *embedding space*. An autoencoder is comprised of two parts: an *encoder*, which transforms the input into its latent representation (*embeddings*), and a *decoder* that reconstructs the input from the obtained embeddings during training (Rumelhart et al., 1986). The encoder and the decoder are trained simultaneously with the objective of minimizing the reconstruction loss, i.e., a measure of how well the decoder can reconstruct the original input from the encoder's output.

Methods like Principal Component Analysis (PCA) dominated the field before autoencoders were introduced. Baldi & Hornik (1989) showed that a single layer encoder without a non-linear activation function converges to a global minimum that represents a subspace of the corresponding PCA. Thanks to their non-linearity, autoencoders allow for a more sophisticated and effective feature extraction, which motivates the use of autoencoders compared to PCA for many applications.

When using autoencoders for dimensionality reduction, the information bottleneck represented by the embedding layer (which is smaller than the input) prevents the encoder from learning the identity function. However, some applications can benefit from over-complete representations, i.e. representations that have a higher latent dimension than their original dimension (e.g. in image denoising Xie et al. (2012) or sparse coding Ranzato et al. (2006)), which work with other methods of regularization. Nonetheless, research has shown that even for applications with an information bottleneck, some forms of regularization can lead to more robust latent representations (Rifai et al., 2011b). One of these methods are contractive autoencoders (CAE).

Moreover, unlike traditional autoencoders, which directly output a latent representation, Variational Autoencoders (VAE) follow a stochastic approach by producing a multivariate distribution parameterized by $\mu, \sigma$ in the latent space Kingma & Welling (2013).

### 2.1 CONTRACTIVE AUTOENCODERS

*Contractive Autoencoders* (CAE) contract the input into a lower-dimensional non-linear manifold in a deterministic and analytical way (Rifai et al., 2011b). Being able to learn very stable and robust representations, CAE were proven to be superior to Denoising Autoencoders (DAE) (see Rifai et al. (2011b)) - where the autoencoder is trained to remove noise from the input for robust reconstructions. In order to achieve the contractive effect, CAE regularize by adding a term to the objective function $\mathcal{J}(\cdot, \cdot)$ alongside the reconstruction loss $d(\cdot, \cdot)$: the squared Frobenius norm $\| \cdot \|_F^2$ of the Jacobian $\boldsymbol{J}_f(\boldsymbol{x})$ of the encoder w.r.t. the input $\boldsymbol{x}$. This encourages the encoder to learn representations that are comparatively insensitive to the input (see Sec. 2 Rifai et al. (2011b)). The objective function can be formally expressed as:

$$\mathcal{J}(\theta, \phi) := \sum_{\boldsymbol{x} \in D} (d(\boldsymbol{x}, D_\theta(E_\phi(\boldsymbol{x}))) + \lambda ||\boldsymbol{J}_f(\boldsymbol{x})||_F^2) \qquad (1)$$

where $\lambda$ is used to factorize the strength of the contractive effect, $D_\theta(\cdot)$ is the decoder parameterized by $\theta$ and $E_\phi(\cdot)$ is the encoder parametrized by $\phi$. As shown in Section 5.3 of Rifai et al. (2011b),

CAE effectively learns to be invariant to dimensions orthogonal to the lower-dimensional manifold, while maintaining the necessary variations needed to reconstruct the input as local dimensions along the manifold. Geometrically, the contraction of the input space in a certain direction of the input space is indicated by the corresponding singular value of the Jacobian. Rifai et al. (2011b) show that the number of large singular values is much smaller when using the CAE penalty term, indicating that it helps in characterizing a lower-dimensional manifold near the data points.

This property of CAE improves the robustness of the model to irrelevant variations in the input data, such as noise or slight alterations, ensuring that the learned representations are both stable and meaningful in capturing the essential features of the data. Additionally, Rifai et al. (2011b) showed the ability to contract in the vicinity of the input data using the contraction ratio defined as the ratio of distances between the two points in the input space in comparison to the distance of the encodings in the feature space. This ratio approaches the Frobenius norm of the encoder's Jacobian for infinitesimal variations in the input space.

Rifai et al. (2011a) also proposed higher order regularization, which leads to flatter manifolds and more stable representations. However, the additional computational cost comes with a limited positive effect, which leads us to use the standard version of first order.

Moreover, Rifai et al. (2011b) show that stacking CAEs can improve performance. A stacked CAE is a series of autoencoders where the embeddings of the first autoencoder are further embedded and reconstructed using the second autoencoder, and so on. Thereby, the contractive penalty term of the encoder is calculated in isolation with respect to the other autoencoders.

## 3 RELATED WORK

While research dedicated to embedding generic or entity-based tabular data is relatively scarce, substantial work has been conducted on embedding data from various other domains including text (Muennighoff et al., 2022; Devlin et al., 2018; Zhang et al., 2023), images, and videos (Kingma & Welling, 2013; Higgins et al., 2016; Comas et al., 2020).

### 3.1 TABULAR DATA EMBEDDING

Ucar et al. (2021) propose to learn representations from tabular data by training an autoencoder with feature subsets, while reconstructing the entire input from the representation of the subset during training as a measure of regularization. Using this methodology, they obtained state-of-the-art results when using the representation for classification. However, they did not perform any dimensionality reduction and hence did not make a comparison on reconstruction quality. Huang et al. (2020) applied a Transformer to encode categorical features obtaining only minor improvements over standard autoencoders despite using a significantly more complex architecture when compared to a simple multilayer perceptron (MLP). Gorishniy et al. (2022) found that complex architectures such as Transformers or ResNets (see He et al. (2015)) are not necessary for effective tabular representation learning, which is in line with our results (see Figure 5).

### 3.2 ENTITY EMBEDDING

As outlined in Section 1, our work aims to embed entities stored in tabular data, especially in industrial settings, where entities are usually platform users, customers, or products. While several approaches have been taken to solve this problem, they usually represent only parts of entities. These include Fey et al. (2023) with an attempt to build a framework for deep learning on relational data and various works on embedding proprietary user/customer interaction data for downstream predictions (Wang et al., 2023; Chitsazan et al., 2021; Chamberlain et al., 2017; Mikolov et al., 2013). Fazelnia et al. (2024) propose a high-level framework for embedding all available information of an entity, including using modality-specific encoders before using a standard autoencoder to produce a unified representation that can then be used for a variety of downstream applications. They also observe a general improvement of downstream prediction performance, but limit their work to their specific proprietary use-case in the audio industry.

## 4 METHODOLOGY

In this Section, (1) we outline DEEPCAE to extend the CAE framework to multi-layer settings while preserving the original regularization design, in contrast to stacked CAE approaches; and (2) we describe our general-purpose entity embedding framework that unifies data pre-processing efforts, which we use to evaluate various embedding models, including DEEPCAE.

### 4.1 DEEPCAE: IMPROVEMENTS TO MULTI-LAYER CAE

Contemplating a multi-layer CAE in our benchmark, we analyze related work and find that, to the best of our knowledge, all use stacking (Wu et al., 2019; Aamir et al., 2021; Wang et al., 2020). This includes Rifai et al. (2011b), who originally proposed the CAE. By stacking the encoders, the layer-wise loss calculations are added up, in contrast to the originally proposed formula in Equation 1 by Rifai et al. (2011b). However, we argue that since $||\boldsymbol{J}_f(\boldsymbol{x})||_F^2 \neq \sum_{i=1}^{k} ||\boldsymbol{J}_i(\boldsymbol{x}_{i-1})||_F^2$ with $k$ being the number of encoder layers, simple stacking puts an unnecessary constraint onto the outputs of hidden layers before reaching the bottleneck layer. The unnecessary constraint originates from penalizing each layer separately using the layer's Jacobian. Instead, when calculating the Jacobian for the entire encoder at once, the encoder has a much higher degree of freedom, while still encouraging a small overall derivative that is responsible for the contractive effect. Consequently, we propose using the actual Jacobian $\boldsymbol{J}_f(\boldsymbol{x})$ of the entire encoder, in line with the originally proposed formula.

Calculating the Jacobian for the entire encoder with respect to every input would be computationally intractable already for two layers when using automatic gradient calculation such as the `jacobian` function in Pytorch Paszke et al. (2019)[1]. For a single fully connected layer with a sigmoid activation, Rifai et al. (2011b) proposed to calculate the Frobenius norm of the Jacobian of the encoder as:

$$\|\boldsymbol{J}_f(\boldsymbol{x})\|_F^2 = \sum_{i=1}^{d_h}(\boldsymbol{h}_i(1 - \boldsymbol{h}_i))^2 \sum_{j=1}^{d_x}\boldsymbol{W}_{ij}^2 \tag{2}$$

where $f$ is the encoder, $\boldsymbol{x}$ is its input, $\boldsymbol{h}$ is its activated output of the encoder and $\boldsymbol{W}$ is the weight matrix of the fully connected encoder layer. Rifai et al. (2011b) show that the computational complexity decreases from $O(d_x \times d_h^2)$ to $O(d_x \times d_h)$, where $d_x$ is the input space, and $d_h$ is the dimension of the hidden embedding space. Note that the term in the outer sum is just the squared derivative of the sigmoid activation function.

To ease the processing of negative input values, we use the $\tanh(x)$ activation function and hence exchange the aforementioned derivative for the derivative of the $\tanh(x)$ activation function. The Frobenius norm of the Jacobian then calculates as:

$$\|\boldsymbol{J}_f(\boldsymbol{x})\|_F^2 = \sum_{i=1}^{d_h}(1 - \boldsymbol{h}_i^2)^2 \sum_{j=1}^{d_x}\boldsymbol{W}_{ij}^2. \tag{3}$$

Note that the fact that the above penalty term can be calculated as a double summation of the layer output and the weight matrix makes this calculation highly efficient. However, as this only works for a single-layer encoder, we provide the necessary derivations for the multi-layer setting in the following, ultimately leading to the DEEPCAE. We start by defining the encoder $f$ as a composite function of its layers (including the activation functions):

$$f := l_k \circ l_{k-1} \circ ... \circ l_1 \tag{4}$$

where each layer $l$ consists of a standard fully connected layer and a $\tanh(x)$ activation. In order to obtain the multi-layer encoders derivative (i.e. the Jacobian matrix), we employ the chain rule:

$$\frac{\delta f}{\delta \boldsymbol{x}} = \frac{\delta l_k}{\delta l_{k-1}} \cdot \frac{\delta l_{k-1}}{\delta l_{k-2}} \cdot ... \cdot \frac{\delta l_1}{\delta \boldsymbol{x}}. \tag{5}$$

---

[1]Initial attempts to employ this method revealed substantial computational demands, even for relatively straightforward cases such as MNIST. The training duration, when applied under identical settings as our proposed methodology, was projected to extend over several days rather than mere hours.

Given the presence of a distinct Jacobian matrix for each layer in the model, we can rewrite this as follows:

$$\boldsymbol{J}_f(\boldsymbol{x}) = \boldsymbol{J}_{l_k}(\boldsymbol{x}_{k-1}) \cdot \boldsymbol{J}_{l_{k-1}}(\boldsymbol{x}_{k-2}) \cdot ... \cdot \boldsymbol{J}_{l_1}(\boldsymbol{x}_0) \qquad (6)$$

where $\boldsymbol{x}_0 = \boldsymbol{x}$ is the input to the encoder.

Given Equation 3, the Jacobian of each layer can be expressed as:

$$\boldsymbol{J}_{l_k}(\boldsymbol{x}_{k-1}) = diag(1 - \boldsymbol{x}_k^2) \cdot \boldsymbol{W} \qquad (7)$$

where $diag(1 - \boldsymbol{x}_k^2)$ is a diagonal matrix, where the diagonal elements are constituted by the components of the vector $\boldsymbol{x}_k$, which represents the output of the layer. This is used with Equation 6 to obtain the Jacobian of the entire encoder. The final penalty term is obtained by taking the squared Frobenius norm of the Jacobian $\boldsymbol{J}_f(\boldsymbol{x})$, expressed as $\|\boldsymbol{J}_f(\boldsymbol{x})\|_F^2$. In our experiments, the full loss function uses the Mean Squared Error (MSE) as the reconstruction loss (cf. Equation 1).

Assuming there are $k$ layers, and weights matrices of dimension $d_x \times d_h$, the complexity of stacked CAEs is $O(k \times d_x \times d_h)$, as the Jacobian is calculated for each layer. Instead, in DEEPCAE the computation of the contractive regularization term is driven by the calculation of the Jacobian of the entire encoder with respect to the input (Equation 6): the overall complexity of DEEPCAE is $O(k \times d_x^3)$, due to the multiplication of Jacobian matrices[2]. As such, the complexity of both stacked CAEs and DEEPCAE scales linearly with the number of layers, but scales cubically with the input size for DEEPCAE, making it less efficient than stacked CAEs, which have a quadratic complexity instead.

Finally, we call our proposed method DEEPCAE, i.e., a multi-layer contractive autoencoder where the contractive loss calculation is based on the Jacobian of the entire encoder, in line with the original design by Rifai et al. (2011b). We compare DEEPCAE to a stacked CAE, and observe that DEEPCAE outperforms it (see Section 5.2).

### 4.2 ENTITY EMBEDDING FRAMEWORK

We outline a general-purpose entity embedding framework (see Figure 1) to generate embeddings and evaluate their quality, and use it to compare DEEPCAE to other embedding methods. This framework can serve as inspiration for practitioners to serve multiple downstream applications with the same entity representation, possibly creating different variants by parameterizing input features and embedding models used.

Starting from the raw data characterizing an entity (e.g. customer metadata, metrics, third-party information), we combine and pre-process it to obtain a tabular dataset. In some cases there may exist additional data structures such as text (e.g. with a natural language description) and time series (e.g. customers purchase patterns per month). Those additional modalities are then embedded through specific encoders such as a pre-trained BERT (Devlin et al., 2018) for textual data, and TS2Vec (Yue et al., 2021) for time-series data. The dataset with all combined features is then fed into an autoencoder model to produce the entity embeddings, which are optionally concatenated with the labels of the corresponding problems, and finally used in downstream applications. This worked for us in a proprietary industrial setting. The datasets in the benchmark do not include text or time series data.

To find the embedding model that is best suited on average for general-purpose entity embedding, we benchmark a variety of autoencoders, including the standard linear autoencoder architecture, a contractive autoencoder, a variational autoencoder and a Transformer-based autoencoder. Beyond the mentioned autoencoder variants, we also employ KernelPCA by Schölkopf et al. (1997) as a non-linear baseline.

## 5 EXPERIMENTS & RESULTS

To identify the best general-purpose entity embedding model, we begin with a comprehensive benchmark on 13 different tabular datasets, as detailed in Section 5.1. We first assess whether the embeddings produced by each method broadly capture the original information using their reconstruction

---

[2]Note that $d_x \geq d_h$.

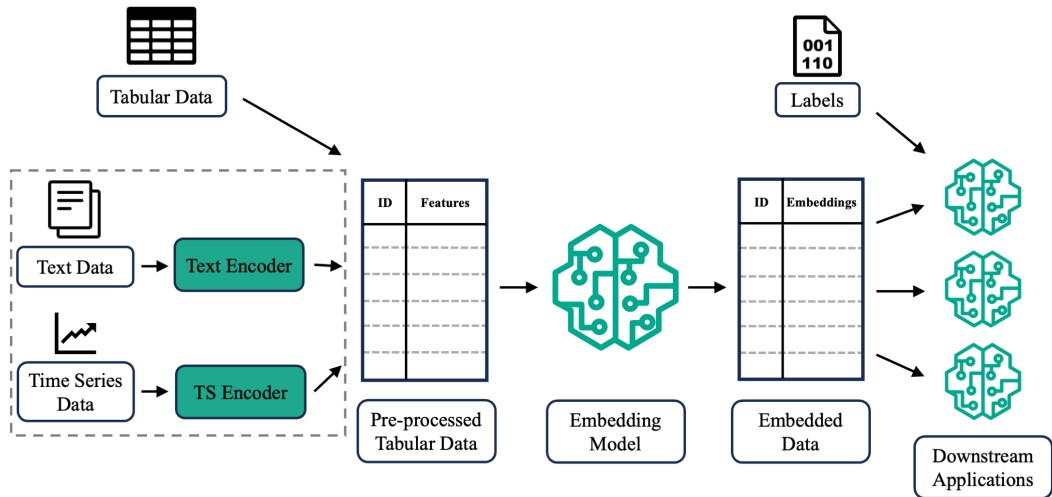

Figure 1: General-purpose embedding framework for multi-modal data and multiple downstream applications. Specific modalities such as text and time-series are embedded via specific methods, and then combined with other tabular data to be fed into the general embedding model. The resulting embedding is then optionally combined with labels, and used by downstream applications.

performance. This is important as general purpose embeddings should be usable for a plethora of downstream applications as introduced in Figure 1. Hence they should capture as much of the available information in the original data as possible and not only that relevant to a specific task. Secondly we directly assess the usability of the embeddings for downstream applications compared the original data. This serves the purpose of testing whether the general purpose embeddings still contain enough use case specific information to make good predictions. Finally, we compare our DEEPCAE to the commonly used STACKEDCAE in Section 5.2.

## 5.1 AUTOENCODERS BENCHMARK

We consider the following embedding models: DEEPCAE, a standard linear autoencoder STAN-DARDAE, a standard autoencoder based on convolutional layers CONVAE, a variational autoen-coder VAE, a Transformer-based autoencoder TRANSFORMERAE, and KERNELPCA (Schölkopf et al., 1997) as a non-linear baseline. We use an adapted version of the Transformer autoencoder in our benchmarks. It uses a transformer block to create a richer representation of the data in en-coder. This representation is then embedded using a small fully connected network projecting into the latent dimension. The decoder consists of a transformer block followed by a linear layer which projects back into the original dimension. All neural network based models are trained with two layers and a compression rate of 50%.

We compare the performances of the models across 13 publicly available tabular datasets listed in Appendix A.1. Before training, commonly used pre-processing methods are applied to convert the data into a fully numerical format. This includes one-hot encoding of numerical features, data type casting, dropping and imputing missing values as well as date conversion to distinct features (year, month, day, weekday). The detailed results for both the reconstruction performance and the downstream performance are provided in Appendix A.3. Details on the model architectures employed are outlined in Appendix A.2.

### 5.1.1 RECONSTRUCTION PERFORMANCE

To evaluate the quality of the resulting embeddings in downstream task-agnostic settings, we pro-pose to assume that reconstruction performance is positively correlated with embedding quality: embeddings that are a rich source of information for the decoder to reconstruct the input will benefit downstream models in task resolution.

We first performed hyperparameter optimization using Asynchronous Sucessive Halving (ASHA)

(Salinas et al., 2022), and then trained each autoencoder and KERNELPCA to evaluate the reconstruction performance on a distinct test set. The reconstruction performance is normalized by that of KERNELPCA to account for the "reconstructability" of a given dataset at a compression rate of 50%.

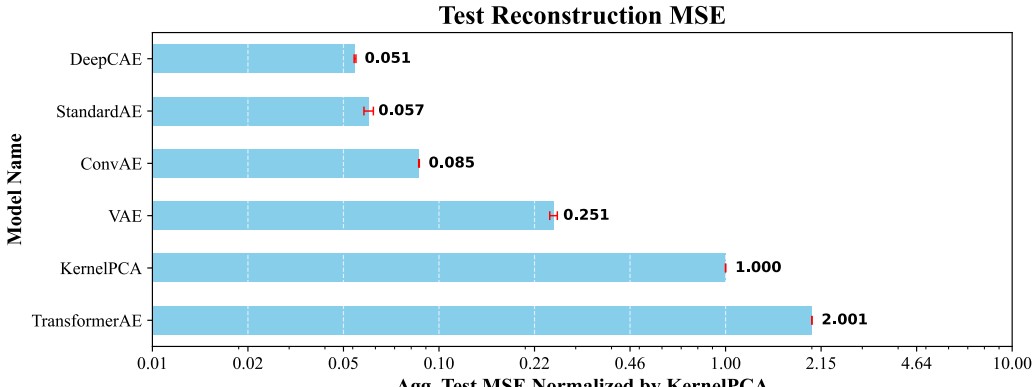

Figure 2: Mean Squared Error (MSE) of reconstruction across 13 datasets (see Appendix A.1), normalized by KernelPCA as the non-linear baseline and aggregated by the geometric mean in logarithmic scale. See STACKEDCAE comparison in Figure 5. Error bars show a 95% confidence interval. **Lower is better.**

From our results in Figure 2, we observe that simpler architectures with a nuanced regularization beyond the information bottleneck (i.e. DEEPCAE) and the vanilla single-layer autoencoder (i.e. STANDARDAE) perform best on average in reconstructing the data. We also observe how the more complex Transformer-based autoencoder performs subpar. We discuss these results in Section 6.

### 5.1.2 DOWNSTREAM PERFORMANCE USING EMBEDDINGS

In pursuit of the best general-purpose entity embedding model, we also assessed the performance of downstream prediction tasks based on the embeddings. This fits well with the workflow of our framework in Figure 1. We trained XGBOOST (Chen & Guestrin, 2016) predictors with the embeddings produced by the set of embedding models described in the previous Section. We then normalized the measured performance by that of an equivalent predictor trained on the raw input data with no loss of information[3]. We use XGBOOST as it is a popular baseline method for classification and regression tasks.

We observe that most embedding variants show competitive performance when compared to a predictor trained on raw data - TRANSFORMERAE and VAE do not. The drop in performance is only marginal for STANDARDAE and DEEPCAE, and is naturally due to the loss of information during the embedding process. It is important to note that in industry or big data settings it is often times infeasible to train the model on all of the data, where we expect increased performance when using embedded data versus either feature subsets or less data for dimensional requirements.

Interestingly, despite the sub-optimal reconstruction performance of KernelPCA (see Figure 2), it outperformed all autoencoders on downstream performance. Moreover, we observe that the decline in downstream performance of most autoencoders is minimal for classification tasks (see Figure 3 - higher is better), while it is more substantial for regression (see Figure 4 - lower is better). These results highlight the capabilities of DEEPCAE and standard autoencoders when it comes to general-purpose entity embedding. We discuss these results in Section 6.

We also applied embeddings on top of a set of customers and tested on internal predictive use cases, including classification and regression. We observe comparative performance or significant improvements on all of the downstream applications where we used customer embeddings to train

---

[3]After the pre-processing described above has been applied.

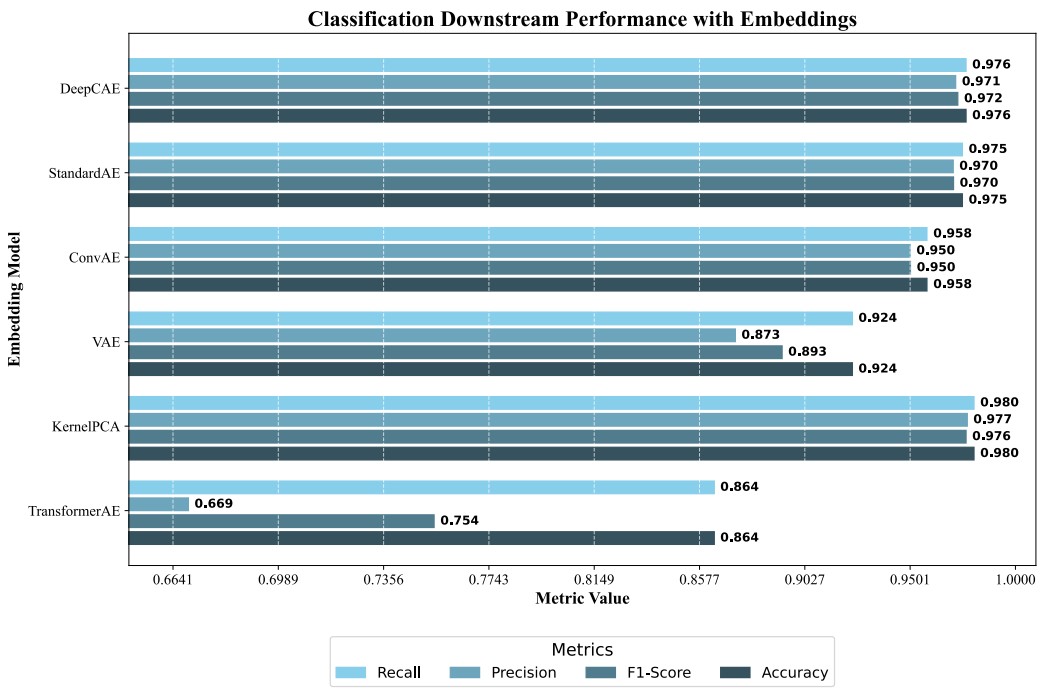

Figure 3: Performance on downstream classification tasks across the classification datasets (see Appendix A.1), normalized by the performance of a predictor trained on the raw data and aggregated by the geometric mean. See STACKEDCAE comparison in Figure 6. **Higher is better.**

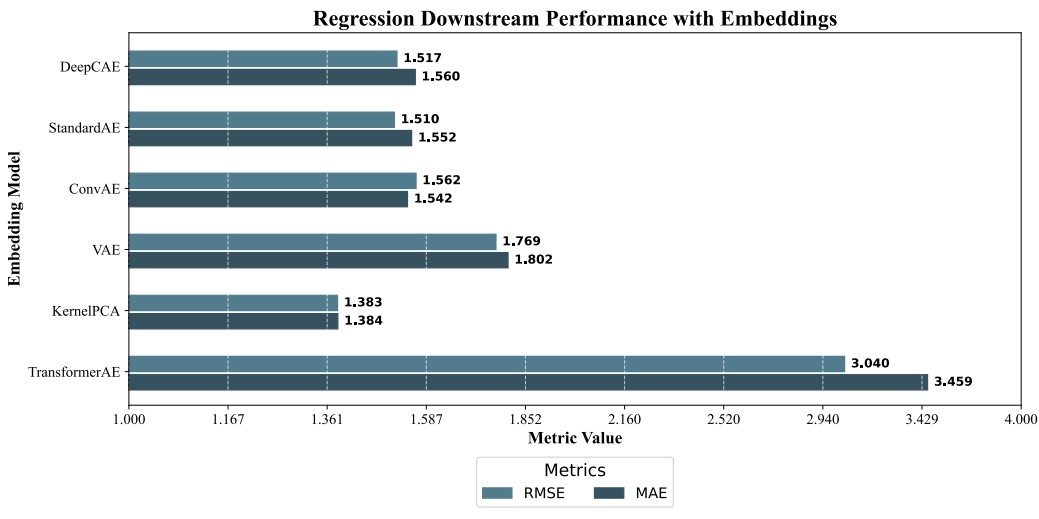

Figure 4: Performance on downstream regression tasks across the regression datasets (see Appendix A.1), normalized by the performance of a predictor trained on the raw data and aggregated by the geometric mean. See STACKEDCAE comparison in Figure 7. **Lower is better.**

the existing application model, despite all downstream models being fed with custom pre-processed and feature engineered data. Performance metrics and problem settings are omitted.

## 5.2 DEEPCAE VS. STACKEDCAE

To provide evidence for our line of argument in Section 4.1 stating that a stacked CAE puts unnecessary constraint onto the hidden layers of the encoder, thereby inhibiting its performance, we benchmark DEEPCAE against a stacked CAE on the aforementioned datasets. We observe that DEEPCAE outperforms STACKEDCAE by $34\%$ in terms of reconstruction performance (see Figure 5), and in terms of downstream performance (see Appendix A.4.2 Figure 6 and Figure 7)

Finally, we compare DEEPCAE and STACKEDCAE on the MNIST hand-written digits dataset as well, in line with CAE's experiments in Rifai et al. (2011b), and observe a $15\%$ improvement over STACKEDCAE.

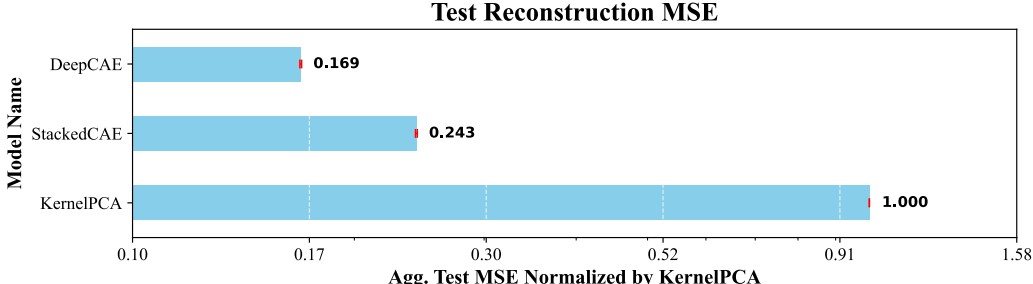

Figure 5: Comparison of stacked CAE and DEEPCAE, normalized by KERNELPCA as a nonlinear baseline and aggregated by the geometric mean in logarithmic scale. Error bars show a 95% confidence interval. **Lower is better.**

The shown improvements in performance through DEEPCAE come at a certain cost: we observe that the average training time across all 13 datasets in our benchmark (cf. Appendix A.1) for DEEPCAE is roughly 6 minutes, while it is only about 3.5 minutes with StackedCAE. The median training time is about 2.5 minutes for both, which confirms that DEEPCAE scales worse than a comparable StackedCAE. The full comparison is given in Appendix A.4.2. For many real-world applications, training times of a few minutes are negligible in the trade-off even for small performance improvements, making DEEPCAE the preferred choice.

## 6 DISCUSSION

### 6.1 DIFFERENCES IN RECONSTRUCTION AND DOWNSTREAM PERFORMANCE

We observe that the reconstruction performance of the various autoencoders is in line with the downstream performance when using the corresponding embeddings (see Figure 2 and Figure 3). Notably, the performance of KERNELPCA surpasses expectations based on the reconstruction benchmark in Figure 2, as it achieves the highest performance in both downstream regression (see Figure 4) and classification benchmarks (see Figure 3).

We hypothesize this difference to be caused by the different incentive of KERNELPCA compared to an autoencoder: KERNELPCA identifies *principal components*, i.e., the directions of maximum variance in the high-dimensional feature space to which the dataset is mapped using a non-linear kernel. By prioritizing principal components, the maximum possible amount of information is retained. While this may not lead to the the best reconstruction in terms of MSE, it explains why KERNELPCA informs downstream predictions so well.

In contrast, the autoencoder's primary incentive for maximizing the information stored in its embedding space is the information bottleneck imposed by the reduced dimensionality and the need to minimize reconstruction loss. However, this approach introduces some slack, as it may deprioritize features with low average magnitude but high variance, which are less penalized by the MSE

reconstruction loss compared to high-magnitude, low-variance features. Future research could explore new loss functions or incorporate feature normalization to address this limitation and improve information retention.

## 6.2 INVERSE RELATIONSHIP BETWEEN MODEL SIZE AND PERFORMANCE

We observe that autoencoder variants with smaller parameter sets exhibit superior performance in both reconstruction and downstream tasks. With reference to the detailed results (see Appendix A.3), we observe that both TRANSFORMERAE and VAE perform comparatively well on larger datasets, but still show lower performance than simpler embedding architectures. Therefore, we hypothesize that these models require large amounts of data for effective training - as is often the case with models that have complex architectures or large parameter sets. While a simple single-layer autoencoder is known to produce transformations similar to those of PCA (Baldi & Hornik, 1989), Transformers and convolutional neural networks (CNNs) augment the input for the extraction of features and representations that are usually of higher dimensionality. While they succeed at representing the input well in a numerical format (e.g. using attention in Transformers (Vaswani et al., 2017)), this may be counterproductive when the objective is to produce a compact, lower-dimensional representation of the input.

## 6.3 SUPERIORITY OF DEEPCAE

Considering all of our benchmarks, and in particular reconstruction performance in Figure 2, we observe DEEPCAE outperforms all of the other embedding models thanks to the regularization introduced by the contractive loss applied across the entire encoder in a multi-layer setting. As explained in detail in Rifai et al. (2011b), this approach helps the model focus on the most relevant aspects of the input while becoming invariant to noise, thereby reducing the risk of overfitting.

## 7 CONCLUSIONS

In this paper, addressing the need for a general-purpose entity embedding model, we proposed DEEPCAE as an extension to the contractive autoencoder CAE in multi-layer settings. We showed that DEEPCAE outperforms all the other tested embedding methods, including a stacked CAE, in terms of both reconstruction quality and downstream performance of classification and regression tasks using the resulting embeddings. The improvement of DEEPCAE over a stacked CAE in terms of reconstruction quality is $34\%$.

Moreover, we outlined a general-purpose entity embedding framework to (a) produce embeddings with different modalities and embedding models, (b) use such embeddings in downstream tasks substituting or augmenting pre-processed data, and (c) evaluate the best embeddings in terms of both reconstruction loss for a task-agnostic purpose and downstream task performance.

Through our experiments, we observed that simpler autoencoders with nuanced regularization, including DEEPCAE, outperform more complex ones, and conclude that these are best suited for robust dimensionality reduction yielding rich embedddings. Furthermore, we argue that the augmentative capabilities of more complex architectures like Transformers and CNNs are not necessarily useful in the production of a compact representation of an entity. Finally, we found that while KERNELPCA is outperformed in terms of reconstruction performance by most of the tested autoencoders, it achieves the best performance in informing downstream predictions with its representations. We concluded that this is due to the inherent limitation of autoencoders, where the focus on minimizing reconstruction loss can introduce slack, leading to suboptimal retention of variance compared to methods explicitly designed to maximize it.

Despite the promising results of DEEPCAE and the successful internal testing on large datasets, the computational complexity introduced by extending the contractive loss to multi-layer settings across the entire encoder may pose scalability challenges for large datasets and high-dimensional data. Future research could focus on addressing these limitations by optimizing the computational complexity of DEEPCAE for scalability. Exploring the integration of new loss functions, including variance maximization, or incorporating more robust feature engineering techniques could further improve the model's ability to retain information.

REPRODUCIBILITY STATEMENT

To ensure the reproducibility of our results, we have submitted the source code necessary to reproduce all experiments as part of the supplementary materials. Detailed descriptions of the data pre-processing steps, assumptions, model architecture, hyperparameters, and training procedure are provided throughout the paper, appendix, and the source code. Details on how to run the source code are provided in the repository's README.md file. We ran each experiment three times and reported the average performance, particularly in terms of reconstruction performance. We ensured the statistical significance of our results by using appropriate evaluation metrics and statistical tests.

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

# A APPENDIX

## A.1 BENCHMARKING DATASETS

Table 1: List of classification datasets used to benchmark the different autoencoder variants.

| Dataset | Description | # Instances | # Features | |
| --- | --- | --- | --- | --- |
| | | | before pre-processing | after pre-processing |
| Adult | Predict if income exceeds $50k/yr. | 48842 | 14 | 107 |
| Bank Marketing | Predict customer behaviour based on data customer metadata. | 45211 | 16 | 46 |
| Churn Modelling | Churn prediction for bank customers. | 10000 | 14 | 2947 |
| Customer Retention Retail | Marketing effects on customer behaviour. | 30801 | 15 | 32 |
| Shoppers | Predict online shoppers purchasing intention. | 12330 | 14 | 20 |
| Students | Predict students' dropout and academic success. | 4424 | 36 | 40 |
| Support2 | Predict patient outcome. | 9105 | 42 | 72 |
| Telco Customer Churn | Predict behavior to retain customers. | 7043 | 21 | 47 |

Table 2: List of regression datasets used to benchmark the different autoencoder variants.

| Dataset | Description | # Instances | # Features | |
| --- | --- | --- | --- | --- |
| | | | before pre-processing | after pre-processing |
| Abalone | Predicting the age of abalone from physical measurements. | 4177 | 8 | 11 |
| AirQuality | Predict PM2.5 amount in Beijing Air. | 43824 | 12 | 15 |
| California Housing Prices | Predict price of houses in California based on property attributes. | 20600 | 10 | 14 |
| Parkinsons | Predict UPDRS scores in Parkinsons patients. | 43824 | 12 | 15 |
| Walmart | Predict weekly store sales in stores. | 6435 | 6 | 6 |

## A.2 AUTOENCODER MODEL ARCHITECTURES AND HYPERPARAMETERS FOR BENCHMARK

Table 3: Detailed architecture and hyperparameters of all models used in the overall benchmark, as well as in the CAE comparison. Hyperparameter tuning was conducted using Asynchronous Successive Halving (ASHA). Model convergence during training was detected using automatic early stopping when no more than 0.2% progress were made in the last 30 epochs.

| Model | Synonym | Architecture | Hyperparameters (fixed) | Hyperparameters (tuned) |
|---|---|---|---|---|
| Standard Autoencoder | StandardAE | Single fully connected linear layer for both encoder and decoder followed by a TanH action respectively. | - | Learning rate, optimized separately for each dataset. |
| Deep Contractive Autoencoder | DEEPCAE | Single fully connected linear layer for both encoder and decoder for the overall benchmark and two layers for the CAE comparison, each followed by a TanH action respectively. | For the CAE comparison the last encoder layer that outputs the final embedding has $d_{input} \cdot compression rate$ and the first encoder layer's output dimension is the average between $d_{input}$ and $d_{embedding}$. | Learning rate, optimized separately for each dataset. Factor $\lambda$ for the contractive loss penalty term with a minimum of 1e-8. |
| Stacked Contractive Autoencoder | StackedCAE | Two CAE that are stacked on top of each other. During training, the contractive loss is calculated for each encoder separately and finally summed up to get the full loss of the StackedCAE. All parts use a fully-connected layer followed by a TanH activation function. | Two CAE are used in stacking, where the last encoder that outputs the final embedding has $d_{input} \cdot compression rate$ and the first encoder's output dimension is the average between $d_{input}$ and $d_{embedding}$. | Learning rate, optimized separately for each dataset. Factor $\lambda$ for the contractive loss penalty term with a minimum of 1e-8. |
| Convolutional Autoencoder | ConvAE | The encoder has two 1D convolutional layers followed by a fully connected layer with a ReLU activation and another fully connected layer with a TanH activation. The decoder has the same in reverse using transposed 1D convolutional layers. | Number of channels: 32 and 64 respectively. | Learning rate, optimized separately for each dataset. |
| Variational Autoencoder | VAE | The encoder consists of 3 1D convolutional layers with ReLU activations followed by two fully connected linear layers with ReLU activations and another linear layer without activation function. The decoder has 3 fully connected linear layers, each followed by a ReLU activation, which are then followed by 3 transposed 1D convolutional layers, with the first two having a ReLU activation and the last one having a TanH activation. | Number of channels: 32 and 64 respectively. | Learning rate, optimized separately for each dataset. |
| Transformer Autoencoder | TransformerAE | The encoder uses a Transformer encoder block followed by a linear layer projecting to the latent dimension. The decoder asymmetrically applies a Transformer encoder block to the embedding, then a linear layer with a TanH activation projecting back to the original input size. | - | Learning rate, optimized separately for each dataset. |

## A.3 Autoencoder Benchmark Results on Tabular Data

This Section contains the reconstruction performance normalized by KernelPCA as a non-linear baseline and the downstream prediction performance using XGBoost normalized by the performance using the original data. The normalization is done to account for the compressability and the predictability in the data respectively in order to focus the comparison on the actual performance of the different methods instead of peculiarities of the datasets. The dimensionality reduction used in all experiments is $\sim 50\%$ (a bit more or less depending for an odd number of features).

### A.3.1 Reconstruction Performance

Table 4: Detailed reconstruction performance measured by MSE on the test set and normalized by the performance of KernelPCA as a non-linear baseline. These results are aggregated from 3 runs each by the geometric mean after normalization with the best architecture bold per dataset.

| Model
Dataset | ConvAE | DeepCAE | JointVAE | PCA | StandardAE | TransformerAE |
|---|---|---|---|---|---|---|
| Abalone | 0.036197 | **0.019002** | 2.008874 | 1.000000 | 0.019761 | 5.544257 |
| Adult | 0.110526 | 0.007267 | 0.033237 | 1.000000 | **0.005416** | 1.082742 |
| AirQuality | **0.180395** | 0.362211 | 0.763212 | 1.000000 | 0.410198 | 22.147542 |
| BankMarketing | **0.013223** | 0.041366 | 0.030352 | 1.000000 | 0.051086 | 2.739541 |
| BlastChar | **0.014618** | 0.051815 | 0.235746 | 1.000000 | 0.032801 | 1.102616 |
| CaliforniaHousing | 0.113429 | 0.045474 | 0.185709 | 1.000000 | **0.043760** | 2.186066 |
| ChurnModelling | 0.375843 | **0.017928** | 1.027224 | 1.000000 | 0.084324 | 1.009939 |
| Parkinsons | 0.050406 | 0.028633 | 0.145312 | 1.000000 | **0.014828** | 1.213129 |
| Shoppers | 0.067997 | **0.043577** | 0.136218 | 1.000000 | 0.050750 | 2.245169 |
| Students | **0.021861** | 0.068385 | 0.080112 | 1.000000 | 0.303356 | 1.277324 |
| Support2 | 0.146305 | 0.052456 | 0.458493 | 1.000000 | **0.049362** | 1.050818 |
| TeaRetail | 0.206707 | **0.077327** | 0.333247 | 1.000000 | 0.078757 | 1.616520 |
| Walmart | 0.919744 | 0.514398 | 0.937297 | 1.000000 | **0.270195** | 1.575301 |

### A.3.2 DOWNSTREAM PREDICTION PERFORMANCE USING XGBOOST (REGRESSION)

Detailed downstream prediction performance on **regression tasks** using XGBoost as a predictor, normalized by the performance of an XGBoost predictor using the original input data as a baseline.

ConvAE

| Dataset | MAE | RMSE |
|---|---|---|
| Abalone | 1.132376 | 1.153945 |
| AirQuality | 1.721003 | 1.681062 |
| CaliforniaHousing | 1.623004 | 1.513682 |
| Parkinsons | 2.173558 | 2.817144 |
| Walmart | 1.267090 | 1.123822 |

DeepCAE

| Dataset | MAE | RMSE |
|---|---|---|
| Abalone | 0.987676 | 0.971155 |
| AirQuality | 1.362350 | 1.382708 |
| CaliforniaHousing | 1.408168 | 1.334347 |
| Parkinsons | 3.988678 | 3.954593 |
| Walmart | 1.222966 | 1.132243 |

**JointVAE**

| Dataset | MAE | RMSE |
|---|---|---|
| Abalone | 1.365242 | 1.302344 |
| AirQuality | 1.666696 | 1.665982 |
| CaliforniaHousing | 2.233703 | 2.006338 |
| Parkinsons | 2.962213 | 3.479520 |
| Walmart | 1.263175 | 1.142287 |

**PCA**

| Dataset | MAE | RMSE |
|---|---|---|
| Abalone | 1.005668 | 1.012301 |
| AirQuality | 1.321127 | 1.310630 |
| CaliforniaHousing | 1.458043 | 1.388065 |
| Parkinsons | 2.220364 | 2.475317 |
| Walmart | 1.178869 | 1.109217 |

**RawData**

| Dataset | MAE | RMSE |
|---|---|---|
| Abalone | 1.000000 | 1.000000 |
| AirQuality | 1.000000 | 1.000000 |
| CaliforniaHousing | 1.000000 | 1.000000 |
| Parkinsons | 1.000000 | 1.000000 |
| Walmart | 1.000000 | 1.000000 |

**StandardAE**

| Dataset | MAE | RMSE |
|---|---|---|
| Abalone | 0.956885 | 0.942567 |
| AirQuality | 1.544251 | 1.536671 |
| CaliforniaHousing | 1.428828 | 1.365434 |
| Parkinsons | 3.524651 | 3.548474 |
| Walmart | 1.209172 | 1.120310 |

**TransformerAE**

| Dataset | MAE | RMSE |
|---|---|---|
| Abalone | 1.468045 | 1.390867 |
| AirQuality | 2.147099 | 2.009472 |
| CaliforniaHousing | 2.985684 | 2.519476 |
| Parkinsons | 42.516824 | 32.692168 |
| Walmart | 1.237763 | 1.128470 |

### A.3.3 DOWNSTREAM PREDICTION PERFORMANCE USING XGBOOST (CLASSIFICATION)

Detailed downstream prediction performance on **classification tasks** using XGBoost as a predictor, normalized by the performance of an XGBoost predictor using the original input data as a baseline.

Table 5: Detailed downstream prediction performance benchmark using embeddings for **ConvAE**

| Dataset | Accuracy | F1-Score | Precision | Recall |
|---|---|---|---|---|
| Adult | 0.929678 | 0.913346 | 0.918707 | 0.929678 |
| BankMarketing | 0.979011 | 0.971462 | 0.968239 | 0.979011 |
| BlastChar | 0.997085 | 0.997553 | 0.998359 | 0.997085 |
| ChurnModelling | 0.942164 | 0.923247 | 0.922109 | 0.942164 |
| Shoppers | 0.988376 | 0.987051 | 0.986287 | 0.988376 |
| Students | 0.921477 | 0.906162 | 0.903675 | 0.921477 |
| Support2 | 0.920583 | 0.920699 | 0.919202 | 0.920583 |
| TeaRetail | 0.989646 | 0.988597 | 0.990347 | 0.989646 |

Table 6: Detailed downstream prediction performance benchmark using embeddings for **DeepCAE**

| Dataset | Accuracy | F1-Score | Precision | Recall |
|---|---|---|---|---|
| Adult | 0.970205 | 0.967522 | 0.967207 | 0.970205 |
| BankMarketing | 0.982305 | 0.975414 | 0.972976 | 0.982305 |
| BlastChar | 1.019614 | 1.015770 | 1.018335 | 1.019614 |
| ChurnModelling | 0.976173 | 0.972559 | 0.972565 | 0.976173 |
| Shoppers | 0.996083 | 0.993851 | 0.993530 | 0.996083 |
| Students | 0.939219 | 0.928102 | 0.922177 | 0.939219 |
| Support2 | 0.930580 | 0.930654 | 0.929096 | 0.930580 |
| TeaRetail | 0.999370 | 0.999127 | 0.999447 | 0.999370 |

Table 7: Detailed downstream prediction performance benchmark using embeddings for **JointVAE**

| Dataset | Accuracy | F1-Score | Precision | Recall |
|---|---|---|---|---|
| Adult | 0.959662 | 0.956317 | 0.955630 | 0.959662 |
| BankMarketing | 0.984841 | 0.975484 | 0.973440 | 0.984841 |
| BlastChar | 0.993764 | 0.986410 | 0.989725 | 0.993764 |
| ChurnModelling | 0.955355 | 0.939221 | 0.939243 | 0.955355 |
| Shoppers | 0.976478 | 0.973899 | 0.972452 | 0.976478 |
| Students | 0.863138 | 0.843025 | 0.836567 | 0.863138 |
| Support2 | 0.710556 | 0.577013 | 0.484665 | 0.710556 |
| TeaRetail | 0.986997 | 0.985919 | 0.987166 | 0.986997 |

Table 8: Detailed downstream prediction performance benchmark using embeddings for **PCA**

| Dataset | Accuracy | F1-Score | Precision | Recall |
|---|---|---|---|---|
| Adult | 0.989686 | 0.987487 | 0.988160 | 0.989686 |
| BankMarketing | 0.987527 | 0.980019 | 0.978381 | 0.987527 |
| BlastChar | 0.986332 | 0.977079 | 0.982530 | 0.986332 |
| ChurnModelling | 0.990686 | 0.984750 | 0.990024 | 0.990686 |
| Shoppers | 0.975901 | 0.974295 | 0.973405 | 0.975901 |
| Students | 0.975969 | 0.971726 | 0.968566 | 0.975969 |
| Support2 | 0.935426 | 0.935604 | 0.934297 | 0.935426 |
| TeaRetail | 1.000669 | 1.000591 | 1.000704 | 1.000669 |

Table 9: Detailed downstream prediction performance benchmark using **RawData**

| Dataset | Accuracy | F1-Score | Precision | Recall |
|---|---|---|---|---|
| Adult | 1.000000 | 1.000000 | 1.000000 | 1.000000 |
| BankMarketing | 1.000000 | 1.000000 | 1.000000 | 1.000000 |
| BlastChar | 1.000000 | 1.000000 | 1.000000 | 1.000000 |
| ChurnModelling | 1.000000 | 1.000000 | 1.000000 | 1.000000 |
| Shoppers | 1.000000 | 1.000000 | 1.000000 | 1.000000 |
| Students | 1.000000 | 1.000000 | 1.000000 | 1.000000 |
| Support2 | 1.000000 | 1.000000 | 1.000000 | 1.000000 |
| TeaRetail | 1.000000 | 1.000000 | 1.000000 | 1.000000 |

Table 10: Detailed downstream prediction performance benchmark using embeddings for **StandardAE**

| Dataset | Accuracy | F1-Score | Precision | Recall |
|---|---|---|---|---|
| Adult | 0.973385 | 0.971705 | 0.970977 | 0.973385 |
| BankMarketing | 0.981957 | 0.974102 | 0.971464 | 0.981957 |
| BlastChar | 1.006581 | 1.006332 | 1.007093 | 1.006581 |
| ChurnModelling | 0.984983 | 0.971822 | 0.978060 | 0.984983 |
| Shoppers | 0.984788 | 0.983711 | 0.982997 | 0.984788 |
| Students | 0.953892 | 0.943782 | 0.939768 | 0.953892 |
| Support2 | 0.916583 | 0.917243 | 0.916844 | 0.916583 |
| TeaRetail | 0.997522 | 0.997200 | 0.997544 | 0.997522 |

Table 11: Detailed downstream prediction performance benchmark using embeddings for **TransformerAE**

| Dataset | Accuracy | F1-Score | Precision | Recall |
|---|---|---|---|---|
| Adult | 0.874236 | 0.758505 | 0.668993 | 0.874236 |
| BankMarketing | 0.976538 | 0.921295 | 0.870823 | 0.976538 |
| BlastChar | 0.953328 | 0.823488 | 0.723387 | 0.953328 |
| ChurnModelling | 0.918418 | 0.821588 | 0.732518 | 0.918418 |
| Shoppers | 0.927442 | 0.848716 | 0.781538 | 0.927442 |
| Students | 0.766732 | 0.621802 | 0.528587 | 0.766732 |
| Support2 | 0.705916 | 0.571017 | 0.478356 | 0.705916 |
| TeaRetail | 0.826057 | 0.732774 | 0.658180 | 0.826057 |

## A.4 CAE COMPARISON: ADDITIONAL BENCHMARKS

This subsection provides additional detail on the comparison of a stacked CAE to DEEPCAE including a comparison on the MNIST dataset (on which the CAE was benchmarked against other autoencoder variants in the paper where it was proposed (Rifai et al., 2011a)). This subsection also provides details on hyperparameters and downstream performance for the datasets used in our main benchmark as listed in Appendix A.1. Beyond that, we provide training times for DEEPCAE, StackedCAE, StandardAE and KernelPCA to discuss the training-cost to performance trade-off that should be considered when using DEEPCAE in a production setting.

### A.4.1 CONTRACTIVE AUTOENCODER BENCHMARK ON MNIST

**Hyperparameters** This is the configuration used for both DEEPCAE and the StackedCAE in the comparison on the MNIST dataset.

| Parameter | Value |
|---|---|
| Layer configuration[4] | 1024, 768, 512 |
| Learning rate | $1.7 \times 10^{-4}$ |
| Number of training epochs | 70 |

Table 12: Model configuration and training parameters for both DEEPCAE and StackedCAE when tested on MNIST.

The factor of the squared Frobenius norm of the Jacobian was customized for the stacked and non-stacked implementation to match roughly for a comparable amount of regularization.

**CAE comparison on downstream performance** CAE comparison on MNIST In advance of this comparison, optimal hyperparameters (see Appendix A.4.1) were determined by automatic hyperparameter optimization using a Bayesian Optimizer by Salinas et al. (2022).

Table 13: MSE between the reconstruction and the original input on the MNIST dataset. Best performance is underlined, our DEEPCAE model is **in bold**.

| Model | Test error | Training error |
|---|---|---|
| **DEEPCAE** | $\underline{1.09\mathrm{e}{-3}}$ $_{\pm 2.28\mathrm{e}{-11}}$ | $\underline{1.08\mathrm{e}{-3}}$ $_{\pm 4.82\mathrm{e}{-13}}$ |
| StackedCAE | $1.28\mathrm{e}{-3}$ $_{\pm 1.42\mathrm{e}{-11}}$ | $1.27\mathrm{e}{-3}$ $_{\pm 2.48\mathrm{e}{-12}}$ |

### A.4.2 CONTRACTIVE AUTOENCODER ON MAIN BENCHMARK

**CAE training time comparison**    We ran the training on all 13 datasets of our main benchmark (cf. Appendix A.1) on an AWS EC2 G6 12xlarge instance and recorded the time it takes to train each model. This instance has 48 vCPUs with 192GB of memory on which KernelPCA was run and 4 Nvidia L4 GPUs (only one of which was used) on which all the other model were trained.

| Model | Mean | Sum | Median |
|-------|------|-----|--------|
| DEEPCAE | 379.898 | 5318.575 | 163.092 |
| PCA | 610.688 | 8549.637 | 43.226 |
| StackedCAE | 209.424 | 2931.939 | 166.812 |
| StandardAE | 144.487 | 2022.811 | 105.452 |

Table 14: Comparison of training runtime in seconds aggregated across datasets.

| Dataset | DeepCAE | PCA | StackedCAE | StandardAE |
|---------|---------|-----|------------|------------|
| Abalone | 64.571754 | 1.574754 | 91.712114 | 57.058129 |
| Adult | 180.479015 | 3454.350542 | 510.017265 | 367.365018 |
| AirQuality | 358.648072 | 483.217805 | 543.527262 | 215.523452 |
| BankMarketing | 270.102090 | 3462.096079 | 265.821518 | 228.812137 |
| BlastChar | 104.311800 | 24.381219 | 165.118584 | 88.736802 |
| CaliforniaHousing | 197.238101 | 57.737437 | 308.244799 | 269.509454 |
| ChurnModelling | 3071.532588 | 76.326727 | 131.427339 | 118.550694 |
| Parkinsons | 322.272196 | 9.473348 | 194.684602 | 92.139630 |
| Shoppers | 145.704254 | 68.812643 | 185.583005 | 142.062517 |
| Students | 113.253082 | 3.664414 | 120.685410 | 49.178837 |
| Support2 | 81.931536 | 28.714640 | 108.195054 | 91.094806 |
| TeaRetail | 245.165447 | 873.916772 | 168.504457 | 194.547937 |
| Walmart | 124.583019 | 4.098899 | 124.324791 | 92.353986 |

Table 15: Comparison of training runtime in seconds for each dataset.

**CAE comparison on downstream performance**    The following plots show the performance of the corresponding XGBoost predictors when using the embeddings from DEEPCAE compared to embeddings from a stacked CAE, KernelPCA and a XGBoost predictors trained on the original data that did not go through any embedding model. The results clearly show how DEEPCAE outperforms a stacked CAE on both classification and regression downstream tasks.

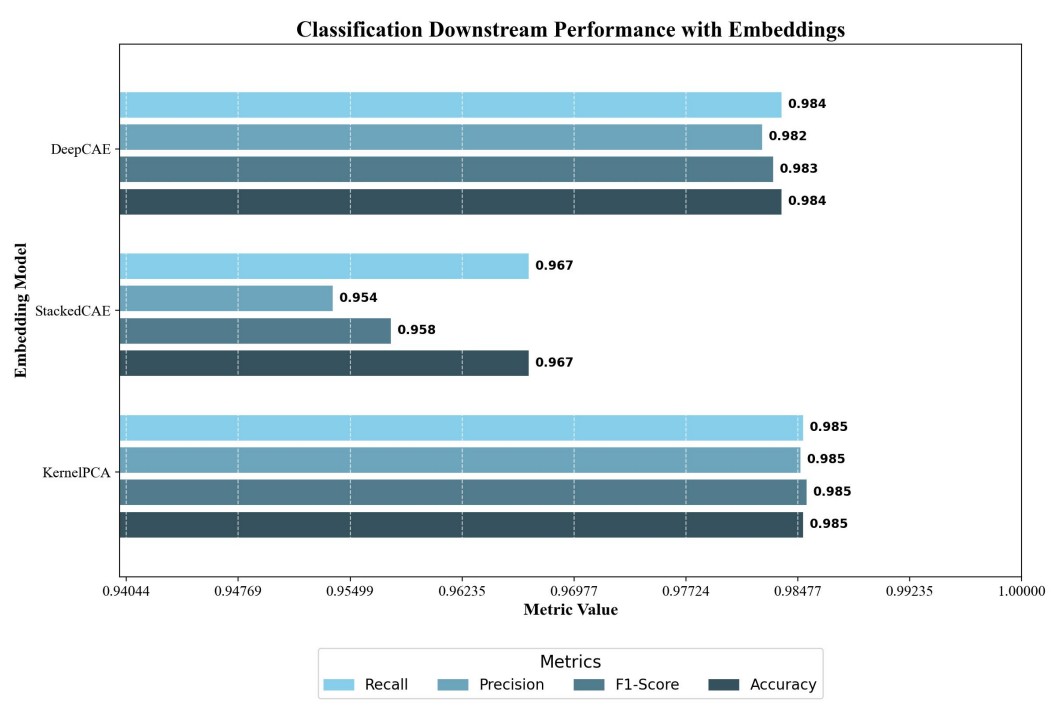

Figure 6: Comparison of stacked CAE and DEEPCAE in terms of downstream classification performance, when using the corresponding embeddings as the information source. Results are normalized by the performance of a predictor trained on the raw data as a baseline and aggregated by the geometric mean. **Higher is better.**

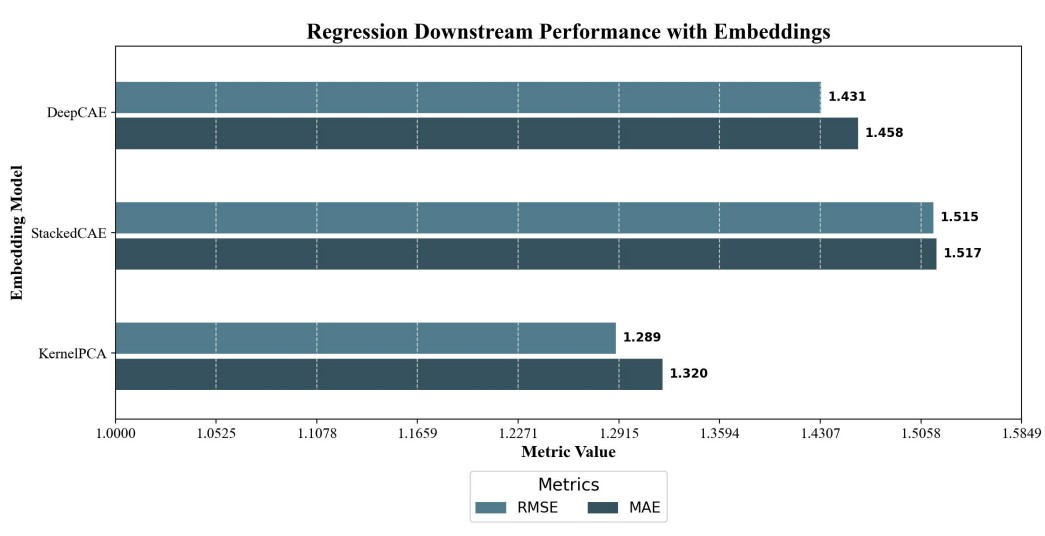

Figure 7: Comparison of stacked CAE and DEEPCAE in terms of downstream regression performance, when using the corresponding embeddings as the information source. Results are normalized by the performance of a predictor trained on the raw data as a baseline and aggregated by the geometric mean. **Lower is better.**

