# OpenReview forum: "Autoencoder-Based General-Purpose Representation Learning for Entity Embedding"
_ICLR.cc/2025/Conference — Submitted to ICLR 2025_

### Official Review · Reviewer_Krnx · 2024-10-22

**Soundness:** 2
**Presentation:** 2
**Contribution:** 2
**Rating:** 3
**Confidence:** 2

**Summary:**

This paper proposes a method called DeepCAE to calculate the regularization term for multi-layer contractive autoencoders and utilizes DeepCAE to power a general-purpose entity embedding framework. The experimental results show that DeepCAE outperformed the baselines.

**Strengths:**

The authors conducted sufficient experiments to validate the effectiveness of their proposed method. They also provide code and data to reproduce the results.

**Weaknesses:**

I think the prominent issue is the writing. 1) It **contains lots of tangential content** which is unnecessary to elaborate on in this paper. For example, in Section 2, I don't see the point of discussing PCA, variational autoencoders, and transformers, as they are not directly part of or foundational to your methodology. 2) The writing **lacks substantial details and is not self-contained**. This work is largely based on the previous CAE work by Rifai et al. (2021b), but the introduction to that work is incomplete, making it difficult to understand the proposed method and its details. Instead, the authors frequently refer readers to the original work. This problem also exists in the experiment, which did not clearly present the experiment settings. 3) **Lacking of logical coherence**. Some important arguments lack evidential support and are not clearly described.

Please see questions below for exact points.

**Questions:**

1. The argument "*Consequently, these representation methods not only are limited, but also fall short or are inapplicable ...*" in line 47-48, do you have evidence to support this claim, such as references or experimental results? Also, what do "*these representation methods*" refer to?
2. In the following paragraph starting at line 50, how are the two contributions related? The authors might consider clarifying that the framework is based on their proposed method.
3. In section 2, what is the purpose of elaborating on PCA, variational autoencoders, and transformers? If they only serve as baselines, a brief introduction in the experimental setup might suffice. Instead, the focus should be on elaborating CAE in this section on its principle and structure. For examples, the steps to encode the input and decode it, the loss function.
4. Eq. (1) lacks detailed description, such as the meaning of its symbols.
5. In line 92-93, "*Thanks to their ability to produce stable and robust embeddings, CAE were proven to be superior to Denoising Autoencoders (DAE)*", the real reason behind *produce stable and robust embeddings* and *superior performance* is missing. The authors should make this argument more well-founded.
6. In the first paragraph of section 4.1, the sentence "*we analyze related work and find that, to the best of our knowledge, all use stacking Wu et al. (2019); Aamir et al. (2021); Wang et al. (2020), including Rifai et al. (2011b), who originally proposed the CAE.*" is poorly written. Additionally, what kind of "stacking" was used? The authors should elaborate on how previous works implemented this.
7. In line 223, how is the conclusion "$O(d_x \times d_h^2)$ to $O(d_x \times d_h)$" derived? I cannot deduce this from Eq. (3) alone. And "*and $d_h$ is the hidden embedding space.*", do you mean $d_h$ is the dimension of the hidden embedding space?
8. The explanation of how Eq.(4) is obtained from Eq. (3) is unclear. The entire inference process from Eq. (3) to Eq. (8) lacks coherence.
9. In line 274, "*such as text and time-series*" what kind of information is the "text" and "time-series" exactly? Could you provide examples?
10. In Figure 1, how exactly do you concatenate the text encoding, TS encoding, and tabular data together?
11. In section 5, what are the experimental settings, such as the number of encoder layers and feature dimensions? Additionally, the dataset statistics should be included in the main paper rather than the appendix, as they are important.
12. How is the Mean Squared Error (MSE) computed? Is this metric conventionally used in previous work?
13. In the first paragraph of section 5.1.2, "*We trained XGBOOST (Chen & Guestrin, 2016) predictors ...*", what is the XGBOOST model, and what exactly is the downstream task?
14. In section 7, line 522-524, "*Furthermore, we argue that the augmentative capabilities of more complex architectures like Transformers and CNNs are not necessarily useful in the production of a compact representation of an entity.*", do you have evidence to support this argument?

---

> ### Author Response · Authors · 2024-11-22
> **General Response**
>
> *References to lines, tables and figures correspond to the new revision, posted together with this answer.*
>
> We thank you for the detailed review. We updated the main text for more focused and self-contained content, as also pointed out by other reviewers. We believe these improvements will greatly benefit the soundness and presentation of the contribution, while clarifying and strengthening the work.
>
> We kindly ask you to review our answers below, where we mention the improvements we made to the paper also in terms of soundness and presentation, and review your scores accordingly. We are available for further clarifications as needed.

---

> ### Author Response · Authors · 2024-11-22
> **Answers to Questions 1**
>
> 1. We agree that the sentence you highlighted needs to be written more clearly and accurately, thank you for pointing it out. We updated the paragraph starting in line 47. \
> Additional context behind our reasoning: \
> With that paragraph, we want to convey that research in the past decade has mostly focused on representation learning of specific modalities, including LSTMs and Transformers for sequential data and with VAEs, GANs and Diffusion models for image data (**“these representation methods”**). Conversely, we could find only little recent research on classical representation learning that could be applied to tabular data during our extensive literature review. However, we argue that classical tabular data is still used in a numerous amount of real-world applications, highlighting the need for a continuation of research in classical representation learning, especially with respect to commonly used entities such as customers and users. This motivates the introduction of DeepCAE, which picks up classical representation learning, extends the CAE framework to multi-layer settings that are feasible under today’s compute resources and outperforms state-of-the-art methods in a tabular setting.
> 2. Your question about the specificity of our framework to DeepCAE is now answered in line 054. There we point out that the general purpose embedding framework is not specific to the DeepCAE method. In the paper it is used to benchmark the different representation learning methods, isolating the performance differences to the difference between architectures. However, as the framework is well suited for operations and real world use we introduce it as a contribution.
> 3. We agree with your point on limiting the preliminaries on the other methods. Following your feedback we updated the paper by (a) shortening these tangential introductions and (b) including more details on CAE in Section 2.1 and Section 4.1. Moreover, we agree that there is a lot to say about CAE, and understanding it deeply requires extensive knowledge of representation learning and linear algebra. While we provide a comprehensive overview, it is hard to provide extensive details while respecting page limits and avoiding repetitions from the original CAE paper. Do you believe there are other key properties of CAE we are missing in Section 2.1?
> 4. We updated the paper to include a more detailed description of Equation 1, as well as a definition for all its symbols.
> 5. How CAE learns stable representations is explained in Section 2.1. We also added some geometrical context for a more complete reasoning. The superiority over denoising autoencoders was shown by Rifai et al. 2011b as cited in our paper. If there are still some aspects around CAE that are not clear to you, please let us know.
> 6. Agreed, good point. We rewrote this sentence to make it easier to read:
> ”we analyze related work and find that, to the best of our knowledge, all use stacking (Wu et al., 2019; Aamir et al., 2021; Wang et al., 2020). This includes Rifai et al. (2011b), who originally proposed the CAE.”
> Following your feedback, we also added details on stacking at the end of Section 2.1. A stacked CAE is a series of autoencoders where the embeddings of the first autoencoder are further embedded and reconstructed using the second autoencoder, and so on. Thereby, the contractive penalty term of the encoder is calculated in isolation with respect to the other autoencoders. We are not aware of any variations of this kind of stacking, and provide an implementation as part of the code repository in the supplementary material.
> 7. Thank you for pointing out the lack of reference here, this conclusion comes from the original CAE paper. We updated the paper with the reference in line 191. On the second point about $d_h$, your understanding is correct that its the dimension of the hidden embedding space. We have clarified this in the main text line 193.
> 8. We updated the derivation of DeepCAE starting in line 195 to make our reasoning more clear and increase coherence.
> 9. In the context of entities such as customers there may be descriptive text data concerning the customer (e.g. a descriptive text from the website of a B2B customer), and time series data containing the purchases of a customer each month. We also updated Section 4.2 with examples for text and time series.
> 10. The numeric embeddings of those time series and text data are appended horizontally as additional columns to the original tabular dataset. The datasets that are used for the experiments that we directly report on in the paper are all tabular data only without free text or time series features to be embedded specifically. We included this in Figure 1 to showcase its applicability and mentioned it worked for us in practice.

---

> ### Author Response · Authors · 2024-11-22
> **Answers to Questions 2**
>
> 11. We updated Section 5 to include details on the experiment settings such as the number of layers (2) and the hidden dimensions (50% of the input dimension) more prominently. We have chosen to include the tables with dataset statistics in Appendix A.1, as they are not directly relevant to the aggregated experimental results discussed in Section 5. Additionally, their inclusion in the main body would occupy significant space, limiting the presentation of other critical details.
> 12. Previous work tends to not compare autoencoder performance based on reconstruction quality at all, hence MSE was not used there. We opted for MSE as a well-known metric for comparing continuous outputs and targets. It is computed over a series of $n$ predictions $\hat y$ and targets $y$ as $MSE(\hat y, y) = \frac 1n \sum ^n_{i=1} (\hat y_i - y_i)^2$.
> 13. The term “downstream applications” in Section 5.1.2. refers to machine learning tasks that use the generated embeddings, instead of the original input, as input to predict a related target, as represented in Figure 1. We decided to use XGBoost for these downstream applications in our benchmark, as it’s not only commonly known and used in the ML community but is also simple and performant. Hence, we assumed no further introduction was needed. However, following your feedback we updated section 5.1.2 by providing additional context on downstream applications and on why we chose XGBoost for downstream tasks.
> 14. Transformers and CNNs are designed to augment the input data in certain ways, which is also why they are so performant in their respective domains (e.g. the attention mechanism in Transformers that correlates different parts of the input sequence to extraction relations between sequence features). However, from our results in Section 5, we observe that both CNN-based and Transformer-based autoencoders do not work well for our setting. From that, we speculate that the augmentative capabilities of these models are not helpful in generating general-purpose entity embeddings. So the evidence of our claim is our results. These are empirical insights on a comprehensive benchmark, and there is room for future work that could try to understand this more in detail. However, since this work is mainly concerned with finding the best entity embedding model, we do not focus on this.

---

> > ### Comment · Reviewer_Krnx · 2024-11-26
> > **Thank the authors for their replies**
> >
> > I have read your replies and the revised paper. As a small suggestion for next time, highlighting the revised text in color would make it easier for readers to track the changes.
> >
> > Honestly, I am not very familiar with CAE model, so the presentation still feels unclear to me from a technical perspective, but I may overlook the significance of DeepCAE' algorithmic contribution.
> >
> > I also agree with reviewer PZNQ that the experimental improvement of DeepCAE is not significant.

---

> > > ### Author Response · Authors · 2024-11-27
> > > **Additional details on CAE, Significance and Contribution**
> > >
> > > Thank you for your suggestion on the coloring, we will take this into account in the future.
> > >
> > > **CAE:** Following the feedback, we added a geometrical explanation to section 2.1 in the paper that introduces CAEs to the reader. We paste the added paragraph here as well for your convenience:
> > >
> > > > Geometrically, the contraction of the input space in a certain direction of the input
> > > space is indicated by the corresponding singular value of the Jacobian. Rifai et al. (2011b) show that the number of large singular values is much smaller when using the CAE penalty term, indicating that it helps in characterizing a lower-dimensional manifold near the data points.
> > > >
> > >
> > > We also added details on each variable in Equation 1 in that section. For an even deeper understanding, we refer the interested reader to Rifai et al. (2011b). However, it is not necessary to understand all the details explained in Rifai et al. (2011b) to understand our results. For simplicity, you could take it as a form of regularization specific to representation learning.
> > >
> > > **Contribution:** Beyond the extension of CAE to our proposed DeepCAE, another contribution of our work is the comprehensive benchmark of representation learning methods for both reconstruction and downstream performance that is to our knowledge and based on our literature review, a novel and far more extensive benchmark beyond anything that existed before in the field.
> > >
> > > **Significance of results:** Please note that we added error bars indicating a 95% confidence interval to the plots showing the results on reconstruction performance (see Figures 2 and 5). Also note that these plots are scaled logarithmically, which makes DeepCAE and StandardAE look closer than they are, at first glance. They are more than 10% apart in terms of reconstruction performance. This clearly shows how DeepCAE generalizes better compared to all other models in the benchmark. If you are still convinced that these results are not significant against the statistical evidence we provide, we kindly ask you to share specific arguments why you think so.

---

> > > > ### Author Response · Authors · 2024-11-30
> > > > **Invitation to Discussion**
> > > >
> > > > We are approaching the end of the discussion period. We did our best to address all questions and weaknesses with data-driven insights to enrich our work. Unfortunately, despite our efforts, we have not yet been able to fully convince you of our proposed framework. If there is anything we can do to further your assessment of the paper and help you better comprehend any necessary details of CAE, please let us know. We will get back to you as soon as possible with additional clarifications. Thank you again for your questions and suggestions that contributed to the enhancement of our work’s presentation.

---

### Official Review · Reviewer_PZNQ · 2024-10-30

**Soundness:** 2
**Presentation:** 1
**Contribution:** 2
**Rating:** 5
**Confidence:** 3

**Summary:**

This paper proposes DeepCAE for learning general-purpose entity embeddings. Although DeepCAE extends from the contractive autoencoder, the authors provide a more effective design in calculating the multi-layered regularization term. Extensive experiments across 13 datasets demonstrate state-of-the-art performance of DeepCAE on reconstruction and ‌downstream‌ prediction tasks.

**Strengths:**

The extension of CAE for multi-layer setting is simple yet effective.

The ‌authors conduct‌ experiments across 13 datasets and ‌cover‌ various types of entities.

The results demonstrate state-of-the-art performance on both reconstruction and ‌downstream‌ prediction.

**Weaknesses:**

The main paper should be self-contained. The authors may overly refer to the ‌original‌ CAE paper.

The motivation of DeepCAE and CAE is not clearly introduced. It is ‌confusing‌ for me why ‌they are designed for tabular data, and how ‌they are connected‌.

In the experimental results, the strengths of DeepCAE are not significant compared with the standard AE.

**Questions:**

Please see the weaknesses.

---

> ### Author Response · Authors · 2024-11-22
>
> *References to lines, tables, figures correspond to the new revision, posted together with this answer.*
>
> Thank you for your review, and for highlighting strengths and weaknesses of our work that we reviewed carefully to improve our work.
>
> We appreciate you highlighting opportunities for improvements on soundness and presentation, and would like to gently ask you to expand on these so that we can make corrections where necessary, and improve presentation aspects such as formatting and readability, especially considering you are fairly confident with your assessment.
>
> We kindly ask you to review our detailed answers below, and review your scores accordingly.
>
> ### Answers to Weaknesses and Questions
>
> [Self-containedness] You rightly noted that our work extends the original CAE framework to multi-layer settings.  Finding the right balance between introducing CAE in detail for a self-contained paper (which often means repeating information from the original CAE paper) and focusing on our extension and its benefits is very difficult and highly subjective. Our approach is to introduce CAE (see section 2.1) with all important properties and refer to the original CAE paper where even the most interested readers can learn all the details. The main motivation for using a CAE is the robustness against small perturbations of the input (i.e., noise) to still capture the essence of the input. Following your feedback, we included a geometrical explanation of the contractive effect to deepen the reader's understanding of contractive autoencoders and thereby make the paper more self-contained.
>
> [Application to tabular data] CAEs are not particularly designed for tabular data. Rather, they are designed to learn representations of any suitable input. Our work is mainly motivated by learning representations of entities stored in tabular format (cf. the paragraph starting in line 47). Hence, we benchmark it on tabular data. Theoretically, it could also work very well for image representation learning. However, there are more tailored approaches for image representation learning (such as VAEs or ViTs), which is why we did not focus on this.
>
> [Motivation of DeepCAE and CAE] Thank you for pointing that out, we clarify the motivation for DeepCAE and CAE in Section 1, line 087 et seq. as well as Section 2.1. line 97 and line 114 et seq. There we point out that the main motivation for using a CAE architecture (both DeepCAE and the original CAE) for embedding tabular data is its robustness against noise in the form of small perturbations of the input while still capturing the essence of the input. In essence, the CAE penalty term is a form of regularization that can improve generalization.
>
> [How are DeepCAE and CAE connected] DeepCAE, StackedCAE, and original CAE all use the contractive term (Frobenius Norm of the encoder’s Jacobian) in the loss computation. However, vanilla CAE only has one layer, while the others have more than one. Moreover, while StackedCAE computes the contractive term for each layer independently, our method DeepCAE takes the more holistic approach of computing the contractive term for the whole network jointly, following CAE’s original design.
>
> We updated the comparison figures with error bars to indicate this in the paper. In the experimental results, the performance difference between DeepCAE and StandardAE is significant under a confidence interval of 95% (cf. Figure 2). Note that Figure 2 is in logarithmic scale, which visually conveys a small improvement between DeepCAE and StandardAE. However, the performance improvement of DeepCAE is about 10%. We specified the logarithmic scale of the x-axis in the paper as well.

---

> ### Author Response · Authors · 2024-11-30
> **Invitation to Discussion**
>
> We are approaching the end of the discussion period. If there is anything beyond what we already addressed, please let us know and we will get back to you as soon as possible with additional clarifications.

---

### Official Review · Reviewer_S38J · 2024-11-02

**Soundness:** 3
**Presentation:** 3
**Contribution:** 3
**Rating:** 8
**Confidence:** 1

**Summary:**

This paper introduces DEEPCAE, a versatile entity embedding framework based on autoencoders. By extending contractive autoencoders (CAE) to a multi-layer structure and preserving the original regularization design, DEEPCAE enhances both reconstruction accuracy and downstream prediction performance for complex entity embeddings. In tests across 13 datasets, DEEPCAE consistently outperformed other autoencoder variants in both reconstruction error and predictive tasks, achieving a 34% reduction in reconstruction error compared to a stacked CAE. This framework offers an efficient, scalable solution for general-purpose entity embeddings across diverse domains, ultimately reducing time spent on feature engineering and boosting model accuracy.

**Strengths:**

This paper introduces DEEPCAE, a multi-layer contractive autoencoder designed for general-purpose entity embedding. By extending the contractive autoencoder framework to multiple layers while preserving regularization, DEEPCAE overcomes limitations seen in stacked CAEs, opening new possibilities for autoencoders with high-dimensional data. The study is thorough, with DEEPCAE evaluated across 13 diverse datasets, showing consistently strong results in both reconstruction and downstream tasks that highlight its effectiveness. The paper is well-organized, with clear derivations and detailed appendices on model architecture and hyperparameters to ensure reproducibility. Overall, DEEPCAE offers an efficient, versatile solution for embedding across domains, reducing feature engineering time and adding practical value for cross-application embeddings in industrial settings.

**Weaknesses:**

DEEPCAE demonstrates impressive results with contractive regularization, but it may not have explored other well-established regularization techniques, like dropout or data augmentation, which are effective in preventing overfitting. It would be beneficial for the authors to consider incorporating these strategies into the DEEPCAE framework. Doing so could enhance the model's robustness, and evaluating their impact on performance in future experiments would provide valuable insights into improving its effectiveness.

**Questions:**

In the existing regularization strategies, have other methods been considered, such as dropout or data augmentation? These techniques have been proven effective in preventing overfitting.

---

> ### Author Response · Authors · 2024-11-22
> **General Response & Answers to Weaknesses and Questions**
>
> Thank you for your review, which highlighted the strengths of our paper and opportunities for improvement.
>
> From your review and summary we are surprised that the confidence level you provided corresponds to not being able to assess our work. We are open to provide further clarifications in case anything is not clear, and to reflect these in our work for further improvement.
>
> ### Answer to Weaknesses and Questions
>
> Regularization methodologies like dropout or data augmentation are indeed effective to further prevent overfitting, however, to allow a fair comparison between model families and architectures, we intentionally omitted such dimensions, leading to clear conclusions. Future work could dive deeper into this aspects, to provide a comprehensive analysis of regularization methodologies in the context of autoencoders.

---

### Official Review · Reviewer_Y2Xz · 2024-11-03

**Soundness:** 4
**Presentation:** 4
**Contribution:** 2
**Rating:** 6
**Confidence:** 3

**Summary:**

In this work the authors extend the Contractive AutoEncoder (CAE) framework for the calculation of the Jacobian
of the entire encoder in the contractive loss from single-layer to multi-layer settings ( DeepCAE ).


Empirically over tabular benchmarks, the authors show DeepCAE can be leveraged in a general purpose embedding
framework where embeddings are feed to XGBoost to obtain gains in reconstruction performance and comparable/slightly better performance
downstream prediction (classification/regression) performance as compared with various AutoEncoders and Transformer baselines
( though not when compared with KernalPCA from a downstream performance perspective ).
They additionally show the noteable reconstruction performance of DeepCAEs compared with Stacked CAEs .

**Strengths:**

The authors show how the CAE framework ( an AE with an additional objective component that is the Frobenius norm of the model with respect to the input ) can be extended to a multi-layer setting in a way that is advantageous when compared with prior extensions to CAEs which worked via stacking.

They do an extensive empirical analysis to discuss reconstruction and downstream accuracy benefits of the method while discussing costs of the method (scaling as the method is cubic with respect to layer size ) and its downstream limitations compared to KernalPCA.

The setting ( encoding tabular data with multi-modal data types  ) is important and they show how its not handled readily or generally by Transformer type architectures.

**Weaknesses:**

1) Time comparisons and particularly error bars needed everywhere ( KernelPCA/StandardAE/DeepCAE ).  For your benchmarks are you all running multiple seeds per each?

The main question ( which the authors have pointed out openly in the paper and for future work ) is if the cost of DeepCAEs is worth the effort?   They’ve shown reconstruction is slightly better, but for tasks its pretty comparable to AE and KernalPCA does better (still an interesting finding ). How important is reconstruction loss really for this setup?  What is the time complexity of KernalPCA and outside of reconstruction loss being subpar are there other reasons to not use it?

2) Are there stronger baselines to compare against both encoding wise and classifier wise (ie, XGBoost vs something else) against if what we care about is tabular performance using embeddings?  The former is the more important of the two and there is a NeurIPS workshop on fusing modalities for tabular data thats in its 3rd edition (https://table-representation-learning.github.io )

3) The general purpose embedding pipeline seems like the standard solution to re-using embeddings for downtsream tasks from vector databases and not particular to DeepCAE?  Is this the case?  If not, it could strengthen the paper to clarify how so if not.

4) It would be interesting to either show experiments on or discuss how DeepCAE does on just image or text data as well to compare its reconstruction and downstream task performance there.  Is there anything in particular that makes this approach specific to tabular data with multi-modal data?  If the method gives performance boosts when encoding image/timeseries/text, it would greatly strengthen the results of the paper and would make incorporating the method.

5) While the background on CAE was very much needed, the sections on VAE and Transformers were probably lesser so ( or could have been pushed into the appendix ) especially since you show effectively they are not nearly as effective.  This space could/should be used for looking more at KernalPCA vs Standard AE vs DeepCAE costs/tradeoffs and potentially other modalities ( point 4)

6) Did you all do experiments against the original CAE?  The paragraph starting at 280 made it seem like you would ( and in the final section you do with StackedCAE), but then in the experiments Convolutional AE is used instead which I wasn't expecting.  I'm assuming this is shown in past work when Stacked CAEs are introduced but having a sense of that as well would be good since its much cheaper computationally than both Stacked and Deep CAEs.

**Questions:**

Q: Did you all empirically assess how much time this k x d^3 adds time-wise compared with just a standard AE?  Its an offline computation, but getting a sense of what that tradeoff comes out to time wise for your datasets would be interesting ( ie, is it that big of a hit in the end since the datasets are all below 45k instances each ).  How does KernalPCA perform?

Q: line 138 should probably cite the 2017 Transformers paper and not the 2023 arxiv one

Q: Is there a reason in particular for using tanh activations in your extension of CAE?  I get it allows for the decomposition shown, but are there other activations or ablations which could have been performed ?

---

> ### Author Response · Authors · 2024-11-22
> **General Response**
>
> *References to lines, tables, and figures correspond to the new revision, posted with this answer.*
>
> Thank you for your detailed review, highlighting the strengths of our paper and opportunities for improvement. We improved the paper where necessary following your questions and suggestions, and provide detailed responses below with references to such improvements in the paper.
>
> We kindly ask you to review our answers below and review your scores accordingly. We are available for further clarifications as needed.

---

> ### Author Response · Authors · 2024-11-22
> **Answer to Weakness 1**
>
> On-Time Comparison, Error Bars, Randomness, Cost-Quality Trade-Off, Reconstruction versus Downstream Measures, KernelPCA:
>
> 1. **DeepCAE Trade-Off**: To assess the trade-off associated with DeepCAE, we added Tables 14 and 15 in Appendix A.4 with a comparison of runtimes in seconds for training DeepCAE, StackedCAE, StandardAE and KernelPCA as well as a brief discussion at the end of the results section at line 460. Also, note that the difference in performance between DeepCAE and StandardAE looks very small in Figure 2 due to the logarithmic scaling, but is more than 10%. For your convenience we also include the paragraph on training time here as well: We observe that average training time across all 13 datasets in our benchmark for DeepCAE is about 6 minutes, while it is only about 3.5 minutes for StackedCAE. The median training time is about 2.5 minutes for both, which confirms that DeepCAE scales worse than a comparable StackedCAE. StandardAE is the fastest to train on average, while PCA takes longer than others on larger datasets. For many real-world applications, training times of a few minutes are negligible in the trade-off even for small performance improvements, making DeepCAE the preferred choice.
>
> 2. **Error Bars**: We added error bars representing the 95% confidence interval for the comparison by reconstruction performance in Figures 2 and 5. The error bars clearly show how DeepCAE outperforms in both comparisons under this 95% confidence interval. In Figure 5, the error bars are very small but non-zero.
> 3. [Random Seeds] Regarding your comment on variation and seeds in our experiments:
>     - For the exact implementation, please refer to our code provided in the supplementary material - we did our best to document it well.
>     - We fixed both the random seeds for splitting the dataset into train and test set, and for validation set for hyperparameter optimization (HPO). This is to keep the split between training and HPO consistent and avoid leakage.
>     - The random seeds for the training and test dataloaders are not fixed to enable Stochastic Gradient Descent (SGD) leading to some variability in the results. For the inference dataloaders (i.e., for embedding generation) the data is not shuffled to isolate the randomness of the downstream modeling for accurate uncertainty reporting.
>     - The seed for model initialization is not fixed to account for parameter initialization in our results. The reported performance in our experiments is the mean of three runs (with different random initialization and shuffled training batches).
>     - The random state of the downstream XGBoost model is not fixed as well to account for different initializations and report downstream modeling uncertainty. The random state is derived from the system clock or another entropy source internally.
>     - As you pointed out, we discuss if the improvements brought by DeepCAE are worth the additional cost and training time at line 460 et seq. at the end of the results section 5. The trade-off between improved embedding quality and cost in terms of time depends on the use case and is hard to be generalized, which is why we do not discuss it further in the paper.
> 4. **Reconstruction Loss**: Reconstruction loss directly quantifies the amount of original information preserved in the embedding. It is important to define a way to evaluate the quality of embeddings for a general-purpose tabular autoencoder, with limited or without downstream tasks to test on. If the decoder can accurately reconstruct the input from the embedding (low reconstruction loss), we can conclude that the embedding retains most or all of the essential information from the original input, making it a good general representation. Following your question, we updated the paper to include this reasoning more explicitly in Table 15 in Appendix A.4.
> 5. **KernelPCA**: KernelPCA has a cubic runtime complexity with respect to the number of data points, making it less suitable for larger datasets (see Table 14). In addition to the weak reconstruction performance, (1) it comes with higher runtime for large datasets, also due to a lack of GPU support, (2) we are not as flexible in encouraging certain properties of the latent space, such as robustness against noise (as possible with DeepCAE) and (3) the non-linear functions can be learned, whereas KernelPCA requires prior definition of the kernel.

---

> ### Author Response · Authors · 2024-11-22
> **Answer to Weakness 2**
>
> Encoding-wise, to the best of our knowledge, we integrated all state-of-the-art comparisons. We thank the reviewer for the opportunity to clarify the downstream task comparison, which was also added in the main text: XGBoost was used as a simple, performant, and widely used proxy for downstream performance. Based on the observed results, we do not expect outcomes that are different from those of other downstream models. Thank you for referencing the NeurIPS workshop. While our work is generally task-agnostic, we see opportunities to further share and/or specify our findings.

---

> ### Author Response · Authors · 2024-11-22
> **Answer to Weakness 3**
>
> Thank you for pointing that out. Our work indeed is not limited to DeepCAE. The end-to-end framework functions as a versatile framework for generating embeddings and solving downstream tasks. By integrating various embedding models into the pipeline while keeping all other components constant, we ensure that comparisons focus solely on differences in model families and architectures. We clarified this in the main text. We hope this will strengthen the impact of our work.

---

> ### Author Response · Authors · 2024-11-22
> **Answer to Weakness 4**
>
> The focus of our work is on classical tabular datasets. DeepCAE is not specifically designed to embed a modality different than tabular such as plain text or images. It is possible to learn representations of flattened images using our method without further modification. This was however not the focus of our work. On the other hand, Transformers exploit the sequential structure in text (syntax and semantics), and CNNs leverage the spatial organization of images. Since CAE’s performance was evaluated on the MNIST dataset, we included a comparison of MNIST in Table 13 in Appendix A.4.2, which makes our work relatable to the original CAE paper and stacked CAE versions.

---

> ### Author Response · Authors · 2024-11-22
> **Answer to Weakness 5**
>
> This point was also highlighted by other reviewers, thank you. We condensed the sections on VAE and Transformers into Preliminaries, and moved the explanation on how we adapted the Transformer architecture for our experiments to the Experiments Section 5.

---

> ### Author Response · Authors · 2024-11-22
> **Answer to Weakness 6**
>
> The original CAE paper (https://icml.cc/2011/papers/455_icmlpaper.pdf) compares the single-layer CAE to StackedCAE in Table 2, and shows CAE with multiple layers outperforms single-layer CAE. Therefore, we took StackedCAE as our baseline.

---

> ### Author Response · Authors · 2024-11-22
> **Answers to Questions**
>
> 1. The difference in complexity between DeepCAE and the other autoencoders leads to noticeable differences in relative runtimes as expected. KernelPCA scales even worse empirically. Find details in Appendix 4.2 and in the aggregated table below (numbers are seconds of training time).
> | Model | Mean | Sum | Median |
> |-------|------|-----|--------|
> | Our Model | 379.898 | 5318.575 | 163.092 |
> | PCA | 610.688 | 8549.637 | 43.226 |
> | StackedCAE | 209.424 | 2931.939 | 166.812 |
> | StandardAE | 144.487 | 2022.811 | 105.452 |
>
> 2. We changed to the 2017 version, thank you for pointing this out.
> 3. The choice of using the $\tanh(x)$ activation function is motivated by the objective of easing the processing of negative input values thanks to the activation function’s output in [-1, 1], differently than ReLU and sigmoid. Moreover, the $\tanh(x)$ activation function also comes with a convenient derivative that is built from the output of the function, which makes derivations in Section 4.1 significantly easier.

---

> > ### Comment · Reviewer_Y2Xz · 2024-11-29
> > **Reviewer reply to rebutall**
> >
> > Thank you to the authors for the clarifications and additions to the paper.
> > The error bars, time comparisons and clarification on the log scale nature of figure 2 improved the strength of the paper in my view, although i think it would behove the authors to fix figure 2 to not be in log scale so as avoid this confusion.  I also think adding mention of the time/tradeoffs in the main paper and then pointing to the time/tradeoffs ( either rounded to the second or at most one decimal point as opposed to three ) in the appendix will also help.  In lieu of this I have updated my score.

---

> > > ### Author Response · Authors · 2024-11-30
> > > **Reply to Reviewer Comment**
> > >
> > > Thank you for your valuable feedback. Thank you once more for your questions and suggestions that contributed to the enhancement of our paper.
> > >
> > > Sadly, the deadline for updating the paper has passed before your reply with new suggestions, such that we cannot include them. However, we want to note that the log scale is now explicitly mentioned in the caption of Figures 2 and also 5. Please also note that we do actually mention the time/performance trade-off in the main paper starting in line 460 at the end of Section 5.

---

### Meta-Review · Area_Chair_aZ2V · 2024-12-17

**Metareview:**

This paper proposes DeepCAE for learning general-purpose entity embeddings. Although DeepCAE extends from the contractive autoencoder, the authors provide a more effective design in calculating the multi-layered regularization term. Empirically over tabular benchmarks, the authors show DeepCAE can be leveraged in a general purpose embedding framework where embeddings are feed to XGBoost to obtain gains in reconstruction performance and comparable/slightly better performance downstream prediction performance as compared with various AutoEncoders and Transformer baselines.They additionally show the noteable reconstruction performance of DeepCAEs compared with Stacked CAEs.

The reviewers generally agree on the sufficient experiments in this work. On the other hand, consistent concerns on (1) self-containing and logical coherence of this paper, (2) quite many presentation issues, (3) significance of the improvement over the original CAE.

**Additional Comments On Reviewer Discussion:**

The aforementioned concerns are still shared among the reviewers.

---

### Decision · Program_Chairs · 2025-01-22

Reject